# The *Escherichia coli* MFS-type transporter genes *yhjE, ydiM, and yfcJ* are required to produce an active $bo_3$ quinol oxidase

**Bahia Khalfaoui-Hassani**[1,2]*, **Crysten E. Blaby-Haas**[3,4], **Andreia Verissimo**[2,5], **Fevzi Daldal**[2]*

**1** Université de Pau et des Pays de l'Adour, E2S UPPA, IPREM, UMR CNRS, Pau, France, **2** Department of Biology, University of Pennsylvania, Philadelphia, PA, United States of America, **3** US Department of Energy Joint Genome Institute, Lawrence Berkeley National Laboratory, Berkeley, CA, United States of America, **4** Lawrence Berkeley National Laboratory, The Molecular Foundry, Berkeley, CA, United States of America, **5** bioMT-Institute for Biomolecular Targeting, Geisel School of Medicine at Dartmouth, Hanover, NH, United States of America

* b.khalfaoui-hassani@univ-pau.fr (BK-H); fdaldal@sas.upenn.edu (FD)

**Data Availability Statement:** All relevant data are within the paper and its Supporting Information files.

## Abstract

Heme-copper oxygen reductases are membrane-bound oligomeric complexes that are integral to prokaryotic and eukaryotic aerobic respiratory chains. Biogenesis of these enzymes is complex and requires coordinated assembly of the subunits and their cofactors. Some of the components are involved in the acquisition and integration of different heme and copper (Cu) cofactors into these terminal oxygen reductases. As such, MFS-type transporters of the CalT family (*e.g.*, CcoA) are required for Cu import and heme-$Cu_B$ center biogenesis of the $cbb_3$-type cytochrome *c* oxidases ($cbb_3$-Cox). However, functionally homologous Cu transporters for similar heme-Cu containing $bo_3$-type quinol oxidases ($bo_3$-Qox) are unknown. Despite the occurrence of multiple MFS-type transporters, orthologs of CcoA are absent in bacteria like *Escherichia coli* that contain $bo_3$-Qox. In this work, we identified a subset of uncharacterized MFS transporters, based on the presence of putative metal-binding residues, as likely candidates for the missing Cu transporter. Using a genetic approach, we tested whether these transporters are involved in the biogenesis of *E. coli* $bo_3$-Qox. When respiratory growth is dependent on $bo_3$-Qox, because of deletion of the *bd*-type Qox enzymes, three candidate genes, *yhjE, ydiM, and yfcJ*, were found to be critical for *E. coli* growth. Radioactive metal uptake assays showed that Δ*ydiM* has a slower $^{64}$Cu uptake, whereas Δ*yhjE* accumulates reduced $^{55}$Fe in the cell, while no similar uptake defect is associated with Δ*ycfJ*. Phylogenomic analyses suggest plausible roles for the YhjE, YdiM, and YfcJ transporters, and overall findings illustrate the diverse roles that the MFS-type transporters play in cellular metal homeostasis and production of active heme-Cu oxygen reductases.

**Funding:** This work was supported by DOE grant DE-FG02-91ER20052 (FD). Work at the Molecular Foundry was supported by the Office of Science, Office of Basic Energy Sciences, of the U.S. Department of Energy under Contract No. DE-AC02-05CH11231 (CEB-H). Work at the U.S. Department of Energy Joint Genome Institute (https://ror.org/04xm1d337), a DOE Office of Science User Facility, is supported by the Office of Science of the U.S. Department of Energy operated under Contract No. DE-AC02-05CH11231 (CEB-H). The funders had no role in study design, data collection and analysis, decision to publish, or preparation of the manuscript.

**Competing interests:** The authors have declared that no competing interests exist.

**Abbreviations:** Cu, copper; *cbb*₃, Cox; *cbb*₃, type cytochrome *c* oxidase; *bo*₃, quinol oxidase; *bo*₃, Qox; HCO, heme-copper oxidases; TM, transmembrane; MFS, major facilitator superfamily; reverse transcription-PCR; RT-PCR, DDM, *n*-Dodecyl β-D-maltoside; UQ1, ubiquinol-1; Na₂S, sodium disulfide; KCN, potassium cyanide.

## Introduction

Respiratory complexes are oligomeric membrane proteins with multiple cofactors, which are widely distributed among prokaryotes and eukaryotes. Their biogenesis is an intricate process involving the insertion of appropriate cofactors into the subunits and assembly of mature subunits into functional enzymes [1–3]. The cytochrome *c* oxidases (Cox) and quinol oxidases (Qox) catalyze the terminal steps of aerobic respiration, which is a four-electron reduction of oxygen to water [4–6]. Both enzymes are multi-heme complexes containing different types of hemes (*a*, *b*, and *c*) [4–6]. However, the two types of oxidases are different with respect to their electron donors as substrates. Cox employs extra-cytoplasmic water-soluble or membrane-attached *c*-type cytochromes, whereas Qox uses lipid-soluble membrane-integral quinones. The *aa*₃-type cytochrome *c* oxidase (*aa*₃-Cox or mitochondrial complex IV) contains two *a*-type (*a* and *a*₃) hemes and also two Cu centers (Cu$_A$ with two Cu atoms and Cu$_B$ with one Cu atom near of the Fe atom of heme *a*₃) [7–9]. The *cbb*₃-type cytochrome *c* oxidase (*cbb*₃-Cox) is exclusively found in prokaryotes, and it contains three *c*-type (*c*$_o$, *c*$_{p1}$ and *c*$_{p2}$) hemes, two *b*-type (*b* and *b*₃) hemes and only one Cu atom near heme *b*₃ iron at the Cu$_B$ center) [10–12]. Some Qox enzymes, like the *Escherichia coli bo*₃-type Qox (*bo*₃-Qox), also contain one Cu atom at their Cu$_B$ center [13] like that of the *aa*₃-Cox or the *cbb*₃-Cox [14,15]. The nature of the cofactors, subunit structures, and electron donors vary among the heme-Cu oxygen reductases but their catalytic Fe-Cu$_B$ centers remain conserved [15,16]. Besides *bo*₃-Qox, *E. coli* has two other *bd*-type Qox enzymes (*bd*-Qox1 and *bd*-Qox2) involved in aerobic respiration, but they contain no Cu atom [6,17,18].

Earlier studies indicated that covalent insertion of the *c*-type hemes to apoproteins is carried out by the cytochrome *c* maturation (*ccm*) systems (*e.g.*, *cbb*₃-Cox) [19]. The Ccm systems operate independently from the insertion of axially coordinated *a*-, *o*-, and *b*-type hemes [2,12]. Coordination of the *b*-type hemes to the apoproteins may be spontaneous, like the soluble four-helical cytochrome *b*₅₆₂ [20]. In other cases, the process might be chaperone-assisted, like the *a*-type hemes of *aa*₃-Cox that rely on the Surf-like (Surf1 or Shy1) proteins [21–24]. In *Paraccocus denitrificans* Surf1 [23] and in *Thermus thermophilus* Surf1q and CbaX [25] are essential to produce active *ba*₃-type quinol oxidases (*ba*₃-Qox), possibly needed for the insertion of the *a*-type hemes. Similarly, the insertion of the *b*-type hemes to the facultative photosynthetic model organism *Rhodobacter capsulatus cbb*₃-Cox requires the CcoS protein [26,27]. While the small protein CydX was proposed to position/stabilize the *b*-type hemes of the *bd*-type quinol oxidase [18,28,29], this process remains unknown in *bo*₃-Qox.

In contrast to the heme groups, Cu insertion into the Cox enzymes has been studied in more detail. In *Rhodobacter* species, the mitochondrial Sco-like proteins [30] SenC or PrrC [31–34], PCuAC-like (PccA) [32,33,35], and Cox11 [36–39] chaperones are involved in this process [39–42]. In the case of *cbb*₃-Cox, Cu is imported by a MFS-type transporter (CcoA) and reduced via a cupric reductase (CcoG) on its way to the cytoplasm [2,43]. Then, Cu is channeled through a specific chaperone (CopZ) and a P$_{1B}$-type transporter (CcoI, CtpA or CopA2) to the periplasmic Sco-like and PCuAC-like chaperones [2,26,44,45], in its way to the Cu$_B$ center of *cbb*₃-Cox [2,46–48]. In *R. capsulatus*, *ccoA* mutants are *cbb*₃-Cox Cu-deficient and unable to import radioactive ⁶⁴Cu [46,47]. This cytoplasmic deficiency can be rescued either by exogenous Cu supplementation, or by deletion of the P$_{1B}$-type Cu exporter CopA, involved in excretion of excess Cu out of the cytoplasm. Remarkably, similar studies in *Rhodobacter sphaeroides* indicated that CcoA is solely dedicated to Cu insertion into the *cbb*₃-Cox and is not required for the similar heme-Cu$_B$ center of *aa*₃-Cox [49]. For the eukaryotic *aa*₃-Cox, Cu located in the mitochondrial intermembrane space is conveyed to the Cu$_A$ center via Cox17 [40–42]. Although no homologue of Cox17 exists in prokaryotes, recently, the

*Bradyrhizobium japonicum* ScoI homologue and the thioredoxin TlpA were shown to metalate *in vitro* the $Cu_A$ center of cognate $aa_3$-Cox [50,51]. Apparently, distinct Cu routes for the biogenesis of similar centers occur in species containing different types of Cox enzymes.

The superfamily of MFS-type transporters belongs to one of the largest groups of secondary active transporters and are exceptionally diverse and ubiquitous to all three kingdoms of living organisms. They selectively transport a wide range of substrates, including sugars, amino acids, peptides, and antibiotics [52]. Despite their structural similarities, members of this superfamily are divided into many families and subfamilies, classified in the IUBMB-approved Transport Classification Database (TCDB, http://www.tcdb.org), based on the diversity of their substrates and their modes of transport (uniporters, symporters, and antiporters). To date, about 105 families of the MFS-type transporters are reported [53], and among them about 28 are classified as Uncharacterized Major Facilitators (UMFs). The CalT subfamily is defined based on their conserved MXXXM and HXXXM motifs [49] and phylogenetic relatedness. They also frequently co-occur with the Cox enzymes [48,49]. The *R. capsulatus* CcoA is the founding member of this subfamily as the first bacterial Cu uptake transporter involved in the biogenesis of the $cbb_3$-Cox [46], and is also the first MFS-type transporter that uses Cu as a substrate [48,49]. Some CcoA-distant members (*i.e.*, the RfnT-like proteins) of the CalT family are also Cu transporters but they do not provide Cu to the $cbb_3$-Cox [48], suggesting that they might play a role in the biogenesis of other cupro-enzymes.

In this work, the role of MFS-type transporters of unknown function (UMFs) in *E. coli* $bo_3$-Qox biogenesis was investigated employing a genetic approach. Using mutants lacking both the *bd*-Qox1 and *bd*-Qox2 enzymes, where the $bo_3$-Qox was the only intact terminal oxidase, the uncharacterized MFS-type transporters YhjE, YdiM, and YfcJ were shown to be required to produce active $bo_3$-Qox to support *E. coli* aerobic respiration. Of these UMFs, YhiE and YdiM affected cellular Fe and Cu homeostasis, respectively, suggesting that MFS-type transporters are required for the biogenesis of different heme-Cu oxygen reductases, possibly as metal or related compound transporters.

## Materials and methods

### Growth conditions, strains and plasmids used

The bacterial strains and plasmids used in this work are described in **S1 Table in S1 File**. All *E. coli* K-12 strains were grown at 37°C on Luria Bertani (LB) enriched or M9 minimal media, supplemented with ampicillin (Amp, 100 μg/ml) and kanamycin (Km, 50 μg/ml), as appropriate. For anaerobic growth, liquid cultures in filled vessels and plates placed in anaerobic jars with $H_2$+$CO_2$ generating gas-packs (Becton, Dickinson and Co., MD) were used. The optical density ($OD_{600}$) of cells in liquid cultures were monitored at 600 nm.

### Kan$^S$ derivatives of the MFS-type transporter mutants

The putative MFS-type transporter mutants ΔsetC, ΔyhjE, ΔyhjX, ΔynfM, ΔydiM, ΔyebQ, ΔyfcJ, ΔaraJ and the ΔcyoB mutant were obtained from the *E. coli* Keio library and were Kan$^R$ [54]. In each case, the kanamycin cassette was removed by introduction of the Flp recombinase carried by the plasmid pEL8 (pCP20) [55], which is Amp$^R$ and temperature sensitive (Ts) for replication. After electroporation, Amp$^R$ mutants harboring pEL8 were grown at 30°C on LB containing ampicillin to allow excision of the *kan* cassette via its FRT sites located adjacent to it. Plates were transferred to 42°C to eliminate the Amp$^R$ provided by pEL18, and the genotypes of the Kan$^S$ and Amp$^S$ colonies were confirmed by PCR using appropriate primers (**S2 Table in S1 File**).

## Construction of the Δbd-Qox1and Δbd-Qox1+Δbd-Qox2 knockout derivatives of selected MFS-type transporter mutants

The Kan$^S$ and Amp$^S$ derivatives of chosen MFS-type transporter mutants were used as recipients to knockout the *bd*-Qox1 and *bd*-Qox2 by P1 transduction. The *Δbd*-Qox1 derivatives of the MFS-type transporter mutants were obtained by using a P1 lysate grown on fresh cultures of the *cydB*:*kan* (Δ*bd*-Qox1) strain, in LB medium supplemented with 0.2% glucose and 5 mM CaCl₂. Before use, the P1 lysates were sterilized with a few drops of chloroform, and the recipient cells were mixed with the P1(*cydB*::*kan*) lysate (at 1:1 v/v ratio), incubated 20 min at 37°C, supplemented with one volume of 1M CaCl₂ and further incubated for 40 min at 37°C in LB medium. The Kan$^R$ (*i.e.*, Δ*bd*-Qox1) transductants were selected on kanamycin containing plates supplemented with 5 mM sodium citrate to chelate Ca$^{++}$ required for P1 reinfection. Following extensive purification, the genotypes of the double (*i.e.*, ΔMFS + Δ*bd*-Qox1) mutants were confirmed by PCR using the primers listed in **S2 Table in S1 File**.

To construct the triple (*i.e.*, ΔMFS + Δ*bd*-Qox1 + Δ*bd*-Qox2) mutants, the Kan$^R$ marker in the *cydB* gene of the ΔMFS + Δ*bd*-Qox1 double mutants was removed using pEL8 as described above. The Kan$^S$ derivatives thus obtained were used to knock out the *bd*-Qox2 by transduction using a P1 lysate obtained by growth on the Δ*appB*::*kan* (*bd*-Qox2) mutant. The triple mutants lacking both the Δ*bd*-Qox1, Δ*bd*-Qox2 and the desired deletion of MFS-type transporter were selected on kanamycin containing plates, and their genotypes confirmed by PCR using the primers listed in **S2 Table in S1 File**.

## RNA isolation and RT-PCR assays

The *E. coli* cells used for RNA isolation and subsequent RT-PCR analyses were grown aerobically at OD₆₀₀ of 0.05, 0.1 (early growth) and 0.15 (late growth), as needed. Prior to RNA extraction, the cultures were washed with sterile water treated with two volumes of "RNAprotect Bacteria Reagent" (Qiagen). The total RNA was extracted using the Qiagen RNeasy mini kit according to the "Enzymatic Lysis of Bacteria" protocol of the manufacturer. 10 μg of total RNA was digested with RNAse-free Dnase I from Qiagen for 25 min at room temperature, followed by overnight precipitation using 20 μl of NaOAc (3M, pH 5.5), 3 μl of glycogen (5mg/ml), and 600 μl ethanol in a final volume of 800 μl. 2 ng of total RNA were used for RT-PCR analyses with OneStep RT-PCR kit from Qiagen using the CyoAQ-F/CyoAQ-R (327 bp amplicon), CyoBQ-F2/CyoBQ-R3 (322 bp amplicon), CyoCQ-F/CyoCQ-R (344 bp amplicon), and CyoDQ-F/CyoDQ-R (310 bp amplicon) primer pairs (**S2 Table in S1 File**) to reverse transcribe and amplify separately portions of mRNA specific of *cyoA*, *cyoB*, *cyoC*, and *cyoD*, respectively. The RrsA-F1 and RrsA-R1 primers were used as an internal control to reverse transcribe and amplify a 100 bp long portion of the 16S ribosomal mRNA. DNA contamination was checked using the master mix containing the heat-inactivated reverse transcriptase (95°C, 15 min) prior to the RT-PCR analyses. The amplified products were separated using 2% agarose gel, and their intensities estimated using ImageJ software (NIH).

## Reduced-minus-oxidized optical difference spectra

To monitor the presence of *bo₃*-Qox in appropriate *E. coli* mutants, optical spectra of *n*-dodecyl β-D-maltoside (DDM)-solubilized membranes from cells grown aerobically at OD₆₀₀ of 0.1 were recorded at room temperature between the 500 and 600 nm using a Varian Cary 50 UV-visible spectrophotometer. DDM-solubilized membrane fractions (final concentration of 5 mg/mL) were prepared in 25 mM Tris-HCl pH 7.0, 150 mM NaCl and 1 mM 4-benzenesulfonyl fluoride hydrochloride (AEBSF). Reduced *minus* oxidized optical difference spectra were

obtained by subtracting the spectra of samples fully reduced with sodium dithionite from the spectra of samples fully oxidized with potassium ferricyanide to detect the $bo_3$-Qox.

## Determination of the $bo_3$-Qox enzyme activity

The oxygen consumption activity of $bo_3$-Qox was monitored using a Clark-type oxygen electrode (INSTECH, Sys203 model). The cells were grown on LB medium to an $OD_{600}$ of 0.1, washed with 0.1 M potassium phosphate buffer, pH 7.0 and resuspended in the same buffer to a total of $OD_{600}$ of 0.5 per assay. 400 μM of ubiquinol-1 was used as an artificial electron donor in the presence of 5 mM of DTT, and the electrode chamber contained one ml of the assay buffer (0.1 M potassium phosphate, pH 7.0, and 0.05% of DDM) at 30°C. The enzymatic reaction was initiated by adding the cells. When tested for inhibitor sensitivity, cells were incubated with either 10 μM of sodium sulfide ($Na_2S$) or 200 μM of potassium cyanide (KCN) for 2 min prior to addition to the reaction mixture. The μM of oxygen consumed/min/$OD_{600}$ of cells were calculated using the formula: Δmm-Hg x 236/140/min/$OD_{600}$ of cells (140 mm-Hg corresponding to 236 μM of oxygen at 30°C was taken as the maximum of oxygen present in the electrode chamber).

## Radioactive ⁶⁴Cu and ⁵⁵Fe uptake assays using whole cells

Whole cells radioactive ⁶⁴Cu uptake assays were performed according to [47]. The radioactive ⁶⁴Cu ($1.84 \times 10^4$ mCi/μmol specific activity) was obtained from the Mallinckrodt Institute of Radiology, Washington University Medical School. *E. coli* strains were grown at an $OD_{600}$ of 0.1 in 10 ml of LB supplemented with the appropriate antibiotics, centrifuged, washed with the assay buffer (50 mM sodium citrate, pH 6.5 and 5% glucose) and re-suspended in one ml of the same buffer. All cultures were normalized to the same number of total cells ($7.5 \times 10^8$ cells) per 500 μl based on their $OD_{600}$ values. Cells were pre-incubated at 35°C or 0°C for 10 min before the assay, and the uptake activity was initiated by addition of $10^7$ cpm of ⁶⁴Cu, determined immediately before use (half-life of ⁶⁴Cu isotope is ~ 12 h). At each time point, 50 μl of 1 mM $CuCl_2$ and 50 μl of 50 mM EDTA (pH 6.5) were added to an aliquot of 50 μl of assay mixture to stop the uptake reaction, and the samples were placed on ice. At the end of the assay, cells were pelleted, pellets washed twice with 100 μl of ice-cold 50 mM EDTA solution, re-suspended in 1 ml of scintillation liquid, and counted using a scintillation counter (Coulter-Beckman Inc.) with wide open window. The uptake assay with ⁵⁵Fe (1 μmol correspond to 73 mCi/mg specific activity) was performed essentially as described for ⁶⁴Cu, except that 1M sodium ascorbate was added to the ⁵⁵Fe stock solution and incubated for 10 min at room temperature to reduce it prior to the assays. The assays were stopped using 1 mM of $FeSO_4$ instead of $CuCl_2$ and processed as described for the ⁶⁴Cu uptake assays.

## Sequence comparison analyses

The protein sequence similarity networks were constructed using the EFI-EST tool (https://efi.igb.illinois.edu/efi-est/) [56] with an alignment score of 70 (YdiM), 110 (YhjE) or 80 (YfcJ), and nodes were collapsed at a sequence identity of 95% (ydiM) or 100% (YhjE and YfcJ). The networks were visualized with Cytoscape (https://www.cytoscape.org) [57] using the Prefuse Force Directed OpenCL Layout. For phylogenetic analyses, protein sequences were aligned using the CIPRES web portal [58] with MAFFT on XSEDE (v. 7.490) [59], and the IQ-TREE web server for construction of phylogenetic trees under maximum likelihood [60]. Trees were visualized with iTOL [61], and branches with less than 50% bootstrap support were deleted. Gene neighborhoods (a window of 10 genes upstream and downstream of the *ydiM*, *yhjE*, or

*yfcJ* homologues) were retrieved using the EFI-GNT tool (https://efi.igb.illinois.edu/efi-gnt/) [56]. The lists of proteins used for bioinformatic analyses can be found in **S1 Dataset**.

## Statistical analyses

In all cases, at least three independent experiments were performed with at least three technical replicates. The error bars reflect the standard deviation with n indicating the number of independent repeats for each experiment. Statistical analyses were performed using the Student t-test with the wild-type activity as reference, and all p-values (when a phenotype is involved) were $<0.05$ as needed.

## Results

### Search for distant CcoA homologues among the E. coli MFS-type transporters

Homology searches were performed to identify putative CalT family members in *E. coli* that contains $bo_3$-Qox, but lacks $cbb_3$-Cox, to inquire whether the two similar heme-$Cu_B$ center containing enzymes share analogous Cu-uptake pathways. Although CalT homologues are readily identified in species belonging to the Gammaproteobacteria [48,49], including *Pseudomonas aeruginosa*, *Shewanella pealeana*, and *Vibrio* species, none are found in the *Enterobacteriaceae*, including *E. coli* (EcoCyc, https://ecocyc.org). Currently there are about 70 ORFs annotated as an "MFS-type transporter" in the genomes of various *E. coli* strains, and about 28 of them have an unknown function (*i.e.*, UMFs). None of these UMFs contains the conserved hallmark (membrane-integral Cu-binding motifs MXXXM and HXXXM) of the CalT family members [11]. This observation suggested that cytoplasmic import of Cu inserted to the *E. coli* $bo_3$-Qox $Cu_B$ center might be delivered by a CalT-unrelated transporter(s), like the *R. sphaeroides* $aa_3$-Cox [49]. However, this suggestion did not exclude whether any one of the UMFs could be involved in $bo_3$-Qox production. Consequently, these UMFs were scrutinized by aligning their amino acid sequences with that of the canonical CalT member (*i.e.*, *R. capsulatus* CcoA) and the occurrence of potential metal binding amino acid residues, like Cys, Met and His [62] (*Supplementary Materials*, **S1 Fig**). This search yielded eight candidates, *yfcJ*, *yhjX*, *yebQ*, *ynfM*, *ydiM*, *yhjE*, *araJ*, and *setC* that were studied further.

### MFS-type transporters that affect the $bo_3$-Qox supported respiration in E. coli

*E. coli* contains three distinct terminal respiratory oxidases, the $bo_3$-Qox, $bd$-Qox-1 and $bd$-Qox-2 that convert oxygen to water during respiration. The $bo_3$-*Qox* is the major enzyme when oxygen concentration is high in the growth media, whereas $bd$-Qox-1 becomes predominant when the oxygen level is low [63,64]. Simultaneous absence of these enzymes renders *E. coli* defective for respiration. However, under certain conditions such as carbon and phosphate starvation, a third $O_2$ reductase, the $bd$-Qox-2 encoded by *appBCX*, could be induced [65,66]. The occurrence of suppressor mutations that turn on the $bd$-Qox-2 is frequent, and this event readily overcomes the respiratory defect of a double mutant lacking both $bo_3$-Cox and $bd$-Qox-1 [67]. Hence, assessing the role, if any, of the UMFs in the production of an active $bo_3$-Cox requires an *E. coli* strain lacking both the $bd$-Qox-1 and $bd$-Qox-2 enzymes. Such a double mutant renders the aerobic respiratory growth of *E. coli* exclusively dependent on the activity of $bo_3$-Qox. Thus, the double deletion $\Delta bd$-*Qox1* + $\Delta bd$-*Qox2* strain (strain BF24 with an active $bo_3$-Qox) and the triple $\Delta bo_3$-*Qox* + $\Delta bd$-*Qox1* + $\Delta bd$-*Qox2* mutant (strain BF17 with an inactive $bo_3$-Qox) were constructed as positive and negative controls for $bo_3$-Qox activity,

## Aerobic conditions

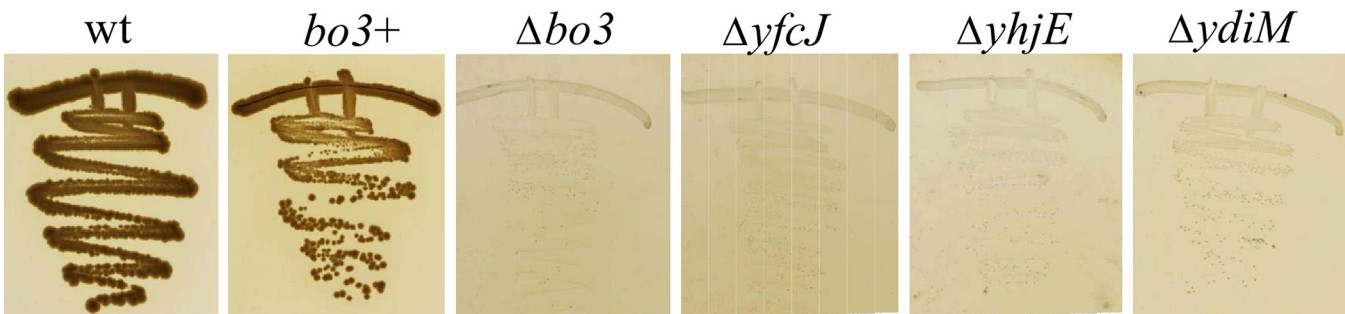

## Anaerobic conditions

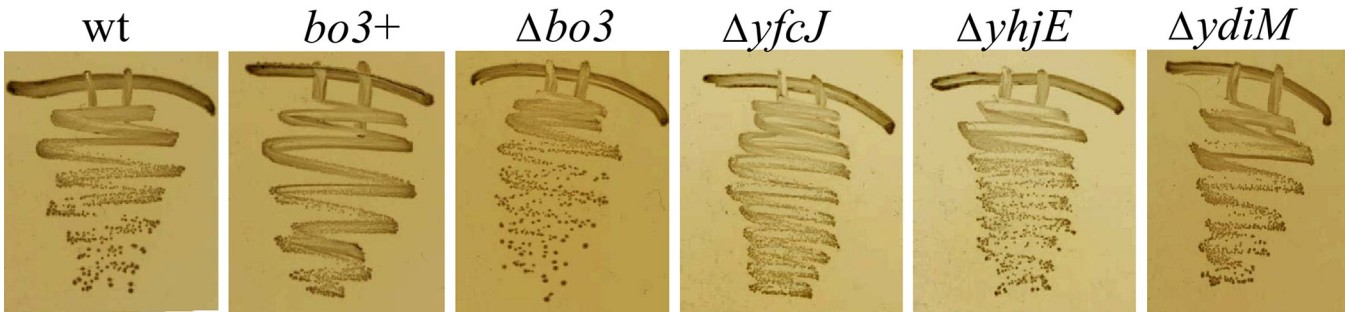

**Fig 1. Growth phenotype of Δ*yfcJ*, Δ*yhjE*, and Δ*ydiM* mutants.** The strains were grown on plates under either aerobic (top row) or anaerobic (bottow row) conditions. The parental strain is the *E. coli* K-12 (BW25113) used to generate the Keo library (**Materials and Methods**). In the $bo_3^+$ strain $bo_3$-Qox is the only functional quinol oxidase, whereas all quinol oxidases are absent in Δ$bo_3$ strain, which does not grow under aerobic conditions. The MFS-type transporters mutants Δ*yfcJ*, Δ*yhjE*, and Δ*ydiM* lack *bd*-Qox1 and *bd*-Qox2 quinol oxidases, and unlike the $bo_3^+$ strain show poor aerobic growth phenotype like the Δ$bo_3$ mutant $bo_3$-Qox.

respectively, using the *E. coli* K-12 Keio collection library [54] (**Materials and Methods**). The deletion alleles of the desired UMFs, equally originating from the Keio library, were introduced under anaerobic growth conditions on minimal medium into the double deletion Δ*bd-Qox1* + Δ*bd-Qox2* background, and their aerobic respiratory growth phenotypes were determined in both minimal (M9) and enriched (LB) media. For the sake of simplicity, these mutants are referred to as $bo_3^+$ (double mutant Δ*bd-Qox1* + Δ*bd-Qox2* with active $bo_3$-Qox), Δ$bo_3$ (triple mutant Δ$bo_3$-Qox + Δ*bd-Qox1* + Δ*bd-Qox2* with inactive $bo_3$-Qox), and Δ*mfs* (triple mutant with a chosen Δ*mfs* + Δ*bd-Qox1* + Δ*bd-Qox2*, where Δ*mfs* corresponds to Δ*yfcJ*, Δ*yhjX*, Δ*yebQ*, Δ*nfM*, Δ*ydiM*, Δ*yhjE*, Δ*araJ*, or Δ*setC*, as appropriate).

As expected, the $bo_3^+$ (Δ*bd*-Qox1 and Δ*bd*-Qox2) strain grew aerobically, though less vigorously than the wild-type parental *E. coli* K-12 (BW25113) strain (**S1 Table in S1 File**), whereas the Δ$bo_3$ strain showed no appreciable respiratory growth (**Fig 1**, top row). When respiratory growth was dependent solely on $bo_3$-Qox, the MFS-type transporter mutants Δ*setC*, Δ*yhjX*, Δ*ynfM*, Δ*yebQ*, and Δ*araJ* were respiration proficient like the $bo_3^+$ (Δ*bd*-Qox1 and Δ*bd*-Qox2) strain. In contrast, the Δ*yhjE* (BF22), Δ*ydiM* (BF23) and Δ*yfcJ* (BF21) mutants exhibited aerobic growth defect like the Δ$bo_3$ strain while their anaerobic growth were fine (**Fig 1**, top and bottom rows). On aerobic-enriched medium, unlike the remaining Δ*mfs* derivatives or the $bo_3^+$ (Δ*bd*-Qox1 and Δ*bd*-Qox2) strain that can attain an $OD_{600}$ of ~ 4 (with 1 h doubling time), the Δ*yhjE*, Δ*ydiM*, and Δ*yfcJ* mutants and the Δ$bo_3$ strain can reach a maximum $OD_{600}$

of only ~ 0.15 (with ~ 4 h doubling time), indicating that their biomass yields were very low. The growth defect of the Δ*yhjE*, Δ*ydiM* and Δ*yfcJ* mutants in the absence of *bd*-Qox1 and *bd*-Qox2 suggested that these UMFs might be required to produce an active *bo₃*-Qox under aerobic growth conditions.

## Effects of YhjE, YdiM, and YfcJ on the transcription of *bo₃*-Qox

Whether the aerobic growth defect seen in the Δ*yhjE*, Δ*ydiM* and Δ*yfcJ* mutants reflected the lack of transcription of the *cyoABCD* operon encoding the *bo₃*-Qox subunits was tested. As the aerobic growth is needed to produce the *bo₃*-Qox, transcription of *cyoB* gene by RT-PCR was used as a proxy for the *cyoABCDE* operon. Mutant cells grown under aerobic conditions at an $OD_{600}$ of ~ 0.05 and ~ 0.1 (early stage of growth) showed that the *cyoB* transcript was detectable in the Δ*yhjE*, Δ*ydiM*, and Δ*yfcJ* mutants (**Fig 2A**). However, at later growth stages ($OD_{600}$ of ~ 0.15 or above) where cell growth was arrested, the amounts of *cyoB* mRNA greatly decreased, possibly reflecting compromised mRNA transcription or stability upon growth stagnation (**Fig 2B**). Hence, the data indicated that at least at the early stage of growth the absence of *yhjE*, *ydiM* or *yfcJ* did not abolish the transcription of *cyoB*. Similar data were also obtained for the *cyoA*, *cyoC* and *cyoD* genes (**S2 Fig**, left lanes). Note that when these mutants were complemented with the multicopy plasmid pJRHisA overexpressing a wild type *bo₃*-Qox, the *cyoA*, *cyoB*, *cyoC* and *cyoD* transcripts were detectable at all growth stages (**S2 Fig**, right lanes).

## Absence of YhjE or YdiM or YfcJ affects the enzymatic activity of *bo₃*-Qox

The *bo₃*-Qox activities of the Δ*yhjE*, Δ*ydiM* and Δ*yfcJ* mutants (in the Δ*bd*-Qox1 Δ*bd*-Qox2 background) were monitored using whole cells at their early stage of growth (at $OD_{600}$ = 0.1, *i. e.*, *cyoABCDE* transcript is like the parent), using a Clark-type oxygen electrode and ubiquinol-1 (UQ1) as an electron donor (**Materials and Methods**). Under these conditions, the *bo₃*⁺ (Δ*bd*-Qox1 and Δ*bd*-Qox2) strain exhibited ~ 34 μmoles of $O_2$ consumed/min/$OD_{600}$ of cells (referred to as 100%). This activity was inhibited by the addition of 10 μM of the *bo₃*-Qox specific inhibitor $Na_2S$ (to ~ 15%) or 200 μM of Cox or Qox inhibitor KCN (to ~ 17%) (**Table 1**), indicating that the measured activity was specific to *bo₃*-Qox. A mutant lacking *bo₃*-Qox (Δ*bo₃*) had ~ 2% of $O_2$ consumption activity that decreased by one half upon addition of either $Na_2S$ or KCN.

In comparison, the Δ*yhjE*, Δ*ydiM* and Δ*yfcJ* mutants (in the *bd⁻ bo₃*⁺ background) exhibited highly decreased activities corresponding to ~ 8%, 4% and 5% compared to the parental strain, and similarly, these activities were inhibited drastically by the addition of $Na_2S$ or KCN (**Table 1**). The data showed that in the absence of the MFS-type transporters YhjE, YdiM, or YfcJ the *bo₃*-Qox activity was drastically reduced, consequently impairing aerobic growth in the absence of the two *bd*-Qox enzymes. As expected, when the Δ*bo₃* strain carried a plasmid born copy of *cyoABCDE* (pJRhisA) (14, 68) its *bo₃*-Qox activity was restored (~ 118.5% ± 6.12), and the *bo₃*⁺ strain carrying the same plasmid overproduced (~ 132.3% ± 0.31) *bo₃*-Qox activity compared to the *bo₃*⁺ parental wild-type strain [14,68]. Increased *bo₃*-Qox activity was observed in the Δ*yhjE* (87.0% ± 5.01) or Δ*ydiM* (49.11% ± 2.4) or Δ*yfcJ* (~ 91.17% ± 6.96) mutants when they carried the plasmid pJRHisA, and their aerobic growth defects were at least partially palliated, yielding increased enzymatic activities in all cases (**Table 1**). In agreement with the earlier transcription profiles, RT-PCR data also indicated that that the plasmid-borne *cyoABCDE* sustained transcription at later stages of growth ($OD_{600}$ of 0.15 or above) (**S2 Fig**, right side).

# A- Early stage of growth

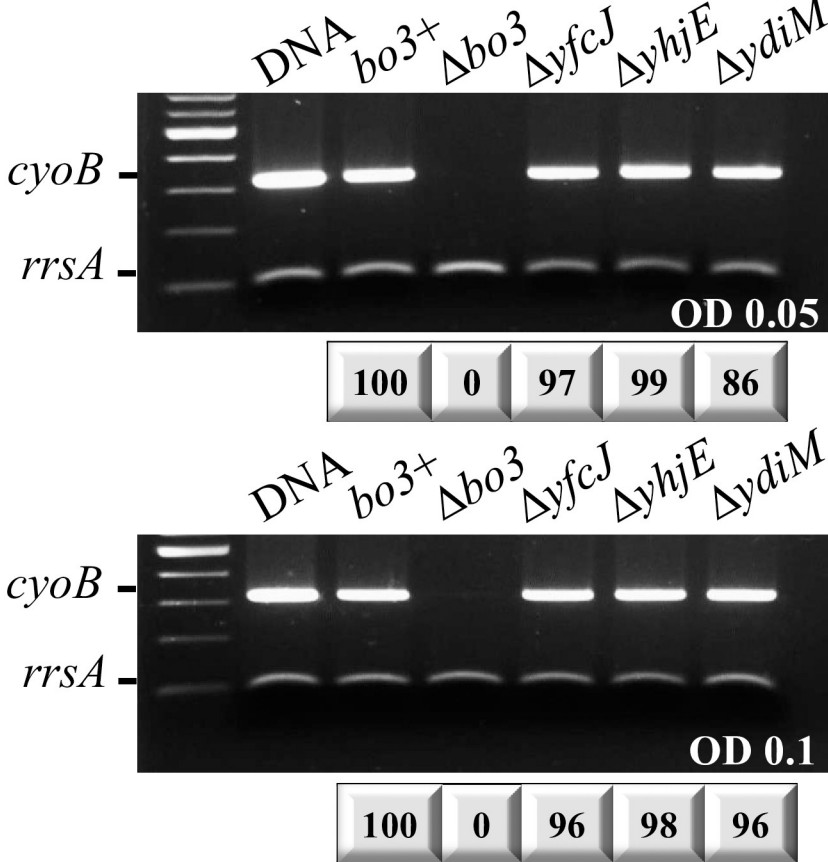

# B- Late stage of growth

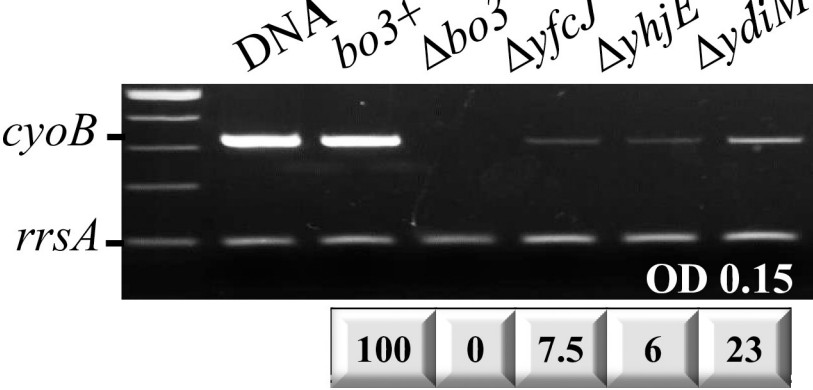

**Fig 2. Effects of the absence of YfcJ, YhjE, and YdiM on the transcription of *bo₃*-Qox.** One step RT-PCR was performed on total RNA extract from *bo₃⁺* strain as well as Δ*bo₃* (*cyoB*), Δ*yfcJ*, Δ*yhjE*, and Δ*ydiM* mutants using the *cyoB* primers to amplify a 322 bp DNA fragment, and the *rrsA* primers to amplify a 100 bp region of the 16S ribosomal mRNA as a control (**Materials** and **Methods**). (**A**) Early stage of growth. The cells were grown under aerobic conditions at $OD_{600}$ of 0.05 (top panel) and 0.1 (bottom panel) where cell division continues. The transcription of *cyoB* was readily detected and showed no difference at both $OD_{600}$. (**B**) Late stage of growth. The cells were grown under aerobic conditions at $OD_{600}$ of 0.15 (maximum $OD_{600}$ reached). The transcripts of *cyoB* gene were barely detectable when the *bo₃*-Qox or the MFS-type transporters YfcJ, YhjE, and YdiM are absent. The numbers below each panel indicate the intensities of the corresponding bands, normalized to that of *rrsA* then compared to that seen with the

$bo_3^+$ strain (taken as 100%). These intensities were determined using ImageJ software (NIH). A control PCR where the reverse transcriptase enzyme was inactivated at 95°C was performed for each total RNA extract to check for DNA contamination. Each experiment is repeated at least three times, and a representative sample is shown for each case.

## Absence of YhjE, YdiM, or YfcJ affects the heme composition of bo₃-Qox

The dithionite-reduced *minus* ferricyanide-oxidized optical difference spectra was obtained using membrane fractions of appropriate *E. coli* mutants grown at an early stage of growth to monitor their *b*- and *o*-type heme compositions. The data indicated that the membranes of the $bo_3^+$ strain or its derivative overproducing $bo_3$-Qox ($bo_3^+$ + pJRhisA), showed a broad band centered at 560 nm with a shoulder at ~ 563 nm (**Fig 3**), characteristic of the presence of the *b* and $o_3$ hemes of $bo_3$-Qox [69]. In the $\Delta bo_3$ strain this band was drastically reduced, consistent with the absence of the $bo_3$-Qox. The remaining small amounts of absorbance likely reflected possible contamination from the abundant periplasmic cyt $b_{562}$. Similarly, the membranes of $\Delta yfcJ$, $\Delta yhjE$, or $\Delta ydiM$ mutants exhibited trace amount of heme *b* and $o_3$ spectra (**Fig 3**), confirming that in the absence of either YfcJ, YhjE, or YdiM the *b*- and *o*-type hemes of $bo_3$-Qox were undetectable despite the presence of *cyoABCD* mRNA, and consistent with the absence of the enzyme activity and defective aerobic growth (**Table 1**). The overall data indicate that the absence of either YhjE, YdiM, or YfcJ abolishes the production of an active $bo_3$-Qox enzyme when this enzyme is expressed from a chromosomal copy, whereas the effect(s) of these UMFs was still apparent but less pronounced when the *cyoABCDE* operon was

**Table 1. The $bo_3$-Qox activity of various strains.**

| Strains | UQ1 | UQ1+ Na₂S | UQ1+ KCN |
|---|---|---|---|
| [a]$bo_3^+$ | [a]100 ± 3.5 | [c]14.7 ± 2.8 | [c]17.0 ± 2.9 |
| $\Delta bo_3$ | 2.2 ± 0.2 | 0.7 ± 0.2 | 0.5 ± 2.3 |
| $\Delta yfcJ$ | 7.9 ± 3.1 | 1.1 ± 2.3 | 1.5 ± 1.1 |
| $\Delta yhjE$ | 4.3 ± 1.1 | 1.8 ± 0.2 | 1.4 ± 0.6 |
| $\Delta ydiM$ | 5.0 ± 0.7 | 0.5 ± 0.6 | 2.7 ± 0.7 |
| [b]$bo_3^+$ + pJRhisA | [b]132.3 ± 0.3 | 22.7 ± 0.7 | 26.3 ± 1.7 |
| $\Delta bo_3$ + pJRhisA | 118.5 ± 6.1 | 13.9 ± 1.8 | 19.0 ± 0.5 |
| $\Delta yfcJ$ + pJRhisA | 91.1 ± 7.0 | 11.5 ± 1.0 | 21. 8 ± 4.3 |
| $\Delta yhjE$ + pJRhisA | 87.0 ± 5.0 | 6.6 ± 2.5 | 15.3 ± 4.4 |
| $\Delta ydiM$ +pJRhisA | 49.1 ± 2.4 | 3.8 ± 0.9 | 8.3 ± 3.7 |

The $bo_3$-Cox activities were measured by monitoring the oxygen consumption activities of whole cells using a Clark-type oxygen electrode. The activities were measured by incubating ubiquinol-1 (UQ1) with sodium dithionate at 30° C prior to adding the cells (see **Materials and Methods**) and all the assays were performed at least three times, with the p values being <0.05 for all mutants.

[a]The parental $bo_3^+$ strain exhibited ~ 34 mmoles of $O_2$ consumed/min/$OD_{600}$ of cells and taken as 100%.

[b]+ pJRhisA refers to the complementation of various mutants with a plasmid harboring $bo_3$-Qox operon [14].

[c]10 μM of sodium disulfide (Na₂S) or 200 μM of potassium cyanide (KCN) were used to inhibit the $bo_3$-Qox activity, as needed.

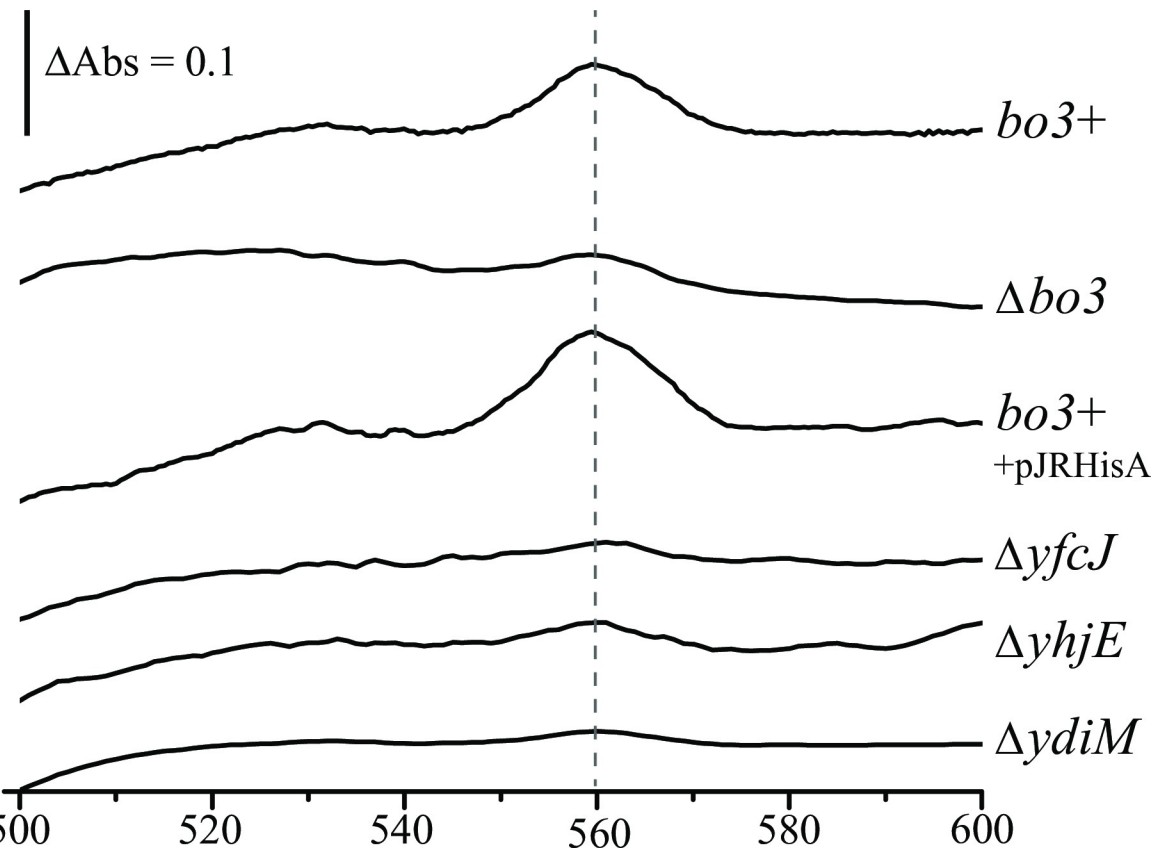

**Fig 3. *b*-type heme compositions of the Δ*yfcJ*, Δ*yhjE*, and Δ*ydiM* mutants.** The reduced *minus* oxidized spectra of the membranes prepared from the *bo₃⁺* strain as well as the Δ*bo₃*, Δ*yfcJ*, Δ*yhjE*, and Δ*ydiM* mutants grown under aerobic condition at an OD₆₀₀ of 0.1. The *bd*-Qox1 and *bd*-Qox2 being absent, the observed broad peak at 560 nm in *bo₃⁺* and the strain overproducing *bo₃*-Qox (*bo₃⁺* + pJRHisA) was taken as corresponding to the hemes *b* and *o₃* of *bo₃*-Qox. This peak is drastically reduced in Δ*bo₃* as well as the Δ*yfcJ*, Δ*yhjE*, and Δ*ydiM* strains. Each experiment is repeated at least three times, and a representative sample is shown for each case.

overexpressed from the multicopy plasmid pJRHisA. Combined with the RT-PCR assays, these results suggest that the negative impact of the Δ*yhjE*, Δ*ydiM*, and Δ*yfcJ* mutants on *bo₃*-Qox gene expression does not fully explain the complete loss of activity of *bo₃*-Qox.

## Cellular $^{64}$Cu or $^{55}$Fe uptake by mutants lacking either YhjE or YdiM or YcfJ

The *E. coli bo₃*-Qox is a heme-Cu containing enzyme, and some members of MFS-type transporters transport Cu (*e.g.*, CalT family members) [46,47,49] or Fe [70] or siderophores [71,72]. Thus, the Δ*yhjE*, Δ*ydiM* and Δ*yfcJ* derivatives of an otherwise wild-type strain (BW25113) were assessed for their abilities to take up radioactive $^{64}$Cu or $^{55}$Fe using whole cells at an early stage of their growth (OD₆₀₀ of 0.1).

In the case of Cu, *E. coli* wild-type cells (BW25113) showed robust, time dependent and temperature sensitive (35°C *versus* 4°C) $^{64}$Cu uptake kinetics (Fig 4A). Under the same conditions, the Δ*yfcJ* and Δ*yhjE* mutants behaved like a wild-type strain in respect to $^{64}$Cu uptake kinetics, indicating that the absence of YfcJ or YhjE had no effect on cellular Cu accumulation. The Δ*cyoB* strain exhibited reduced $^{64}$Cu uptake kinetics, suggesting that in the absence of *bo₃*-Qox, the main cupro-enzyme present in *E. coli*, cellular Cu accumulation decreased, possibly due to Cu homeostasis. Remarkably, the Δ*ydiM* mutant also exhibited slow $^{64}$Cu uptake

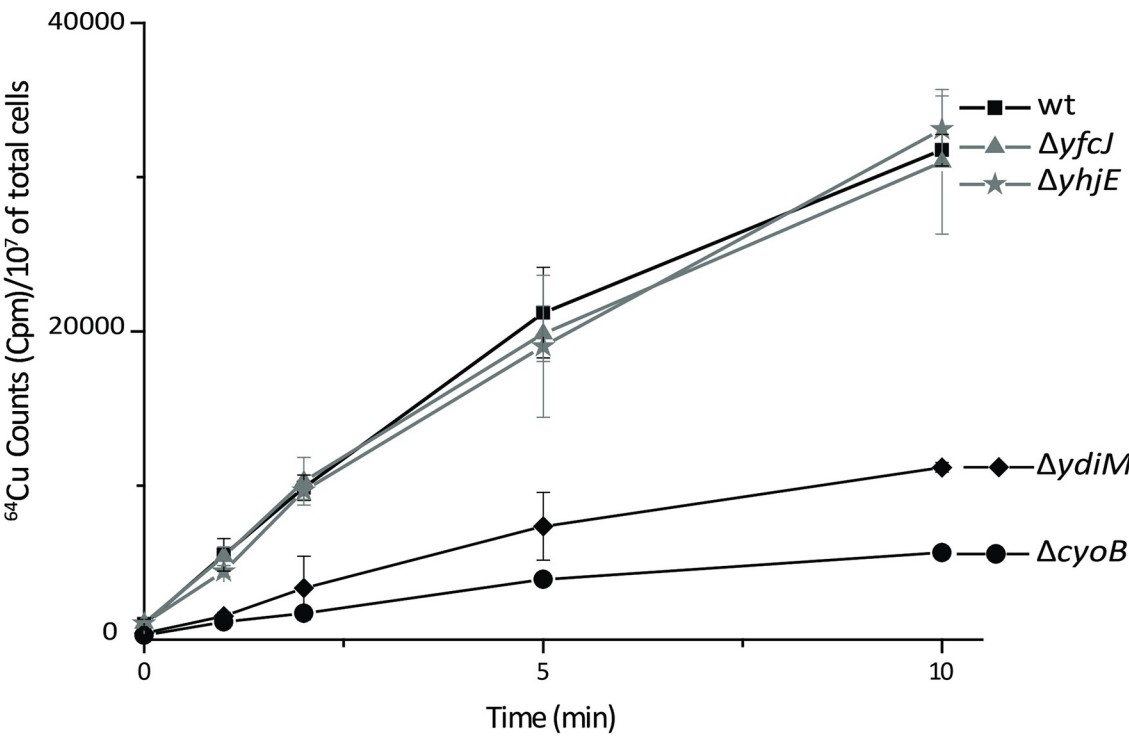

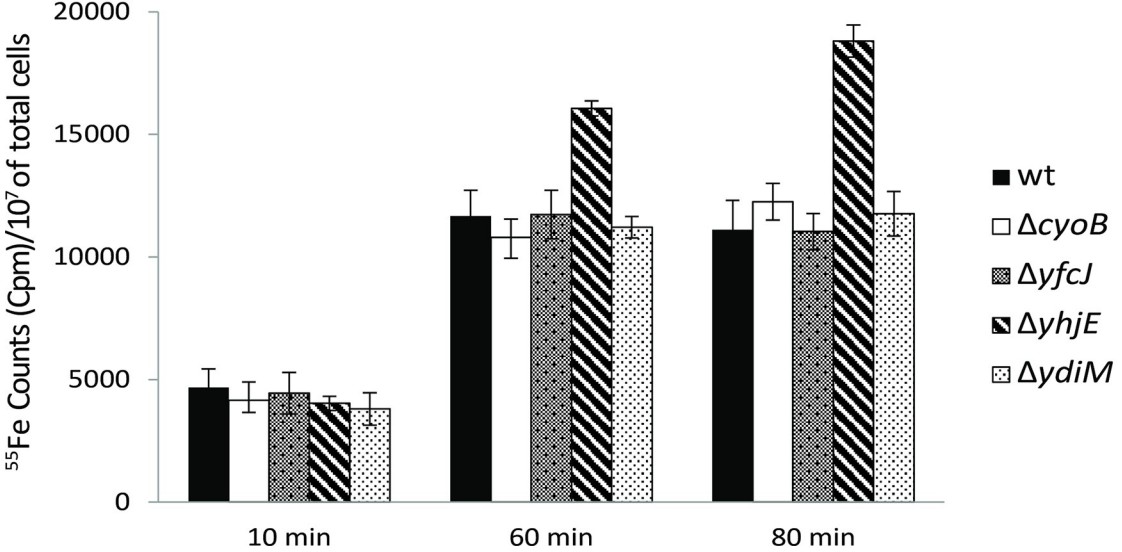

**Fig 4. Whole cell radioactive ⁶⁴Cu and ⁵⁵Fe uptake kinetics of the Δcyo, ΔyfcJ, ΔyhjE, and ΔydiM mutants.** ⁶⁴Cu (top panel) and ⁵⁵Fe (bottom panel) uptake kinetics (**Materials and Methods**) were carried out at 35°C using whole cells grown aerobically until an $OD_{600}$ of 0.1. In each case, the uptake assays were repeated at least three times using at least three independently grown cells, and the p values were $> 0.05$ The ΔydiM and ΔyhjE mutants shows lower ⁶⁴Cu ($p < 0.05$) and higher ⁵⁵Fe ($p < 0.05$) accumulations in cells, respectively.

kinetics (**Fig 4A**) compared with it parental strain, indicating that cellular Cu accumulation decreased. This behavior was reminiscent to that observed with the *R. capsulatus ccoA* mutant that is defective in ⁶⁴Cu uptake [47]. As controls, when the assays were performed at 4° C, all strains showed greatly reduced rates of ⁶⁴Cu uptake.

In the case of Fe, the uptake of $^{55}$Fe-sodium ascorbate (*i.e.*, reduced iron) followed similar kinetics for the wild type and the *DycfJ* and *DydiM* mutants, except the Δ*yhjE* strain that accumulated higher amounts of cellular $^{55}$Fe (**Fig 4B**). Since the assays report the net accumulation of the radioisotope used (*i.e.*, total import *minus* total export during a given incubation period), the data suggested that the Δ*yhjE* was either overactive for import, or deficient for export, of cellular Fe leading to gradual accumulation over the time (**Fig 4B**). As in the case of Cu, when the Fe uptake assays were performed at 4˚ C, very reduced $^{55}$Fe uptake rates were observed. Further, when uptake assays were performed without prior incubation of $^{55}$Fe with sodium ascorbate (*i.e.*, with oxidized form of $^{55}$Fe), then all strains including the Δ*yhjE* exhibited comparable $^{55}$Fe uptake activities. Thus, YhjE affected the transport of reduced, but not oxidized, form of Fe. Overall, whole cells uptake kinetics indicated that the absence of YdiM and YhjE perturbs Cu and Fe homeostasis, respectively, in *E. coli* cells. How the cellular imbalance of Cu or Fe in mutants lacking these two MFS-type transporters is linked to the observed *bo*₃-Qox deficiency and aerobic growth defect, requires further studies.

## YhjE is related to the putative hydroxy-ethyl-thiazol (HET) and other transporters that cluster with bo₃-Qox

YhjE (TC: 2.A.1.6.10) belongs to a large subfamily of the MFS-type transporters with homologues in most major bacterial phyla. In TCDB, YhjE is listed as a metabolite:H+ symporter (MHS) family member, but its metabolite cargo is unknown. YhjE homologues are identified in beta- and alpha-proteobacteria in addition to gamma-proteobacteria, with sequence similarity hits (based on top 1000 blastP hits against UniProt reference proteomes) being predominately related to proteins from the Proteobacteria (49%) and Actinobacteria (43%). A sequence similarity network analysis indicates that of the previously published MFS-type transporters, YhjE is most similar to ThiU [73] (TC 2.A.1.6.12; putative thiazol transporter according to TCDB) (**Fig 5A**) (**S4 Fig**). Based on previous gene clustering and phylogenetic profiling analyses, ThiU is predicted to be a hydroxy-ethyl-thiazole (HET) transporter required for thiamin biosynthesis [73], although this hypothesis has not been tested experimentally. Based on phylogenetic reconstruction, ThiU and YhjE may be paralogs, suggesting that ThiU and YhjE could have separate functions (**Fig 5A**). As an example, the *Haemophilus influenzae* genome encodes a ThiU ortholog (HI_0418) located in the thiamine-related genes cluster, and a YhjE ortholog (HI_0281) next to two genes encoding enzymes involved in menaquinone biosynthesis (2-succinyl-6-hydroxy-2, 4-cyclohexadiene-1-carboxylate synthase/2-oxoglutarate decarboxylase (MenD; HI_0283) and menaquinone-specific isochorismate synthase (MenF; HI_0285). Intriguingly, MenD is a thiamine-dependent protein. (**Fig 5A**). However, this proximity between a gene encoding YhjE-like proteins and a gene encoding MenD is only observed in *Haemophilus* species.

Noticeably, YhjE-like proteins are identified in some Proteobacterial genomes that are encoded by genes next to the *cyoABCD* operons, encoding the structural subunits of *bo*₃-Qox (**Fig 5B**). Except for the gene clusters from Thiotrichales, Chromatiales, and Hyphomicrobiales, where the *yhjE*-like gene is in the same operon with *cyoABCD*, most *yhjE*-like genes are found in the opposite orientation, suggesting that although *yhjE* and *cyoABCD* may not form an operon, still they might be co-regulated (**Fig 5C**). These YhjE-like proteins, although not connected to the main sequence similarity network cluster (*i.e.*, have an $E_{value} > 1E-70$ with any other protein in the main cluster), might be closely related to YhjE in the phylogenetic tree (**Fig 5B**). This observation suggests that the role of YhjE in *bo*₃-Qox function is likely conserved outside of *E. coli*.

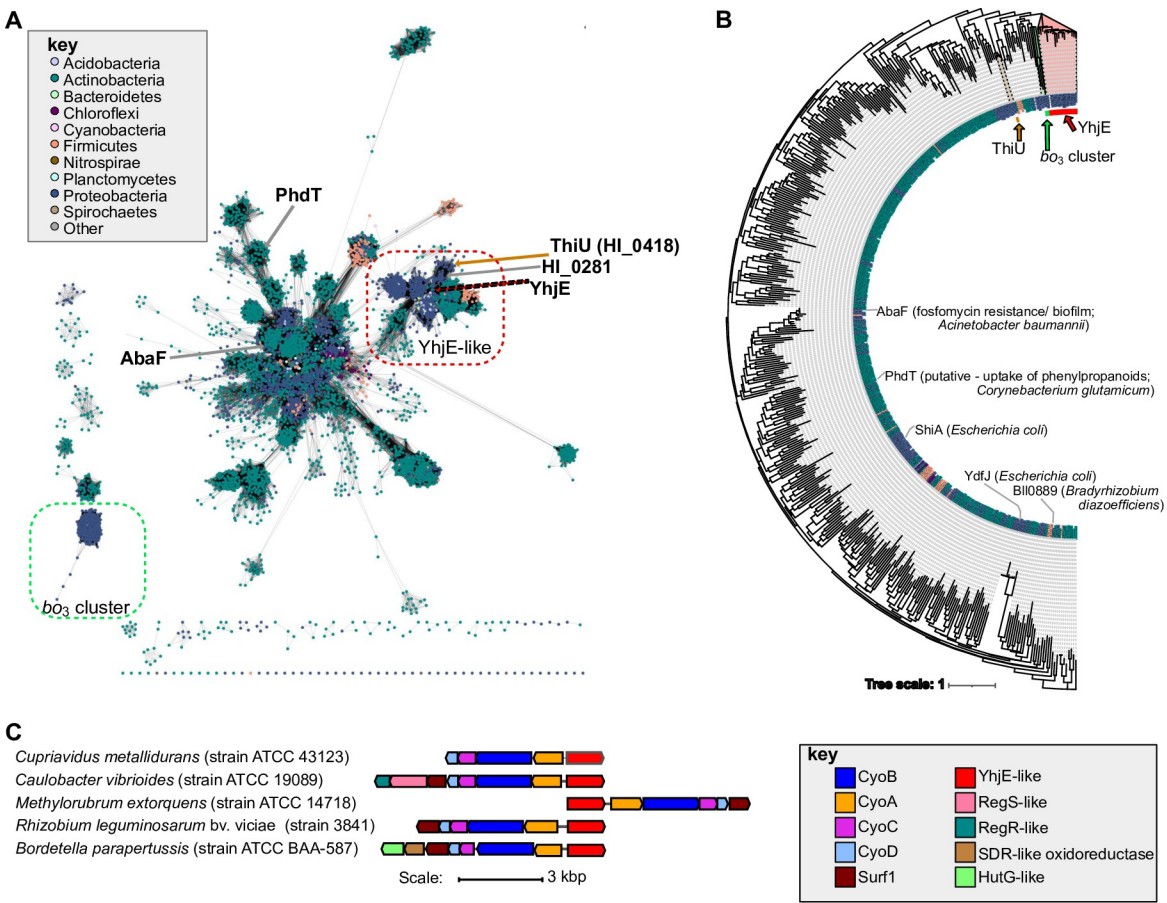

**Fig 5. Sequence similarity analysis of YhjE-like transporters and gene neighborhoods containing its homologues.** (**A**) Sequence similarity network using an alignment score of 110. Sequences for the network were collected by searching against UniRef50 (500 hits) with YhjE and mapping to UniRef90 for network construction. Experimentally characterized proteins or proteins with predicted functions (in italics) based on bioinformatic analyses are labeled. The taxonomic classification of each node is colored according to the key shown on top left. (**B**) iqTREE using edited MAFFT alignment based on UniRef50 sequences, and (**C**) examples of gene neighborhoods containing a YhjE-like MFS transporter with the key defining them located at the bottom right.

## The ydiM-like genes are linked to the shikimate pathway

In the *E. coli* K-12 genome, *ydiM* (TC: 2.A.1.15.12) is located next to its paralog *ydiN* (TC: 2.A.1.15.13), and two other genes encoding two enzymes in the shikimate pathway, *aroD* and *ydiB*, encoding 3-dehydroquinate dehydratase and quinate/shikimate dehydrogenase enzymes, respectively. The shikimate pathway is a major link between carbohydrate metabolism and the biosynthesis of aromatic compounds via chorismate, a precursor of aromatic amino acids phenylalanine, tyrosine, and tryptophan. The cargo of YdiN is not listed in TCDB, but it was previously hypothesized to transport a shikimate by-product [74] based on the genomic context and co-expression data of the *ydiN*, *aroD*, and *ydiB* genes. YdiM is listed as a putative isoprenol exporter due to increase susceptibility that it provides to *E. coli* upon its deletion [75].

Phylogenetic and genomic context analyses of YdiM and YdiN homologues further defined their relationship to the shikimate pathway. The YdiM and the YdiN orthologous group are largely limited to Enterobacterales genomes and not widespread in Proteobacteria. YdiM/YdiN-like proteins are frequently found in Firmicute genomes, represented by YfkL in *Bacillus subtilis* (**Fig 6**). Accordingly, among YdiM homologues (top 1000 blastP hits) ~ 57% are from

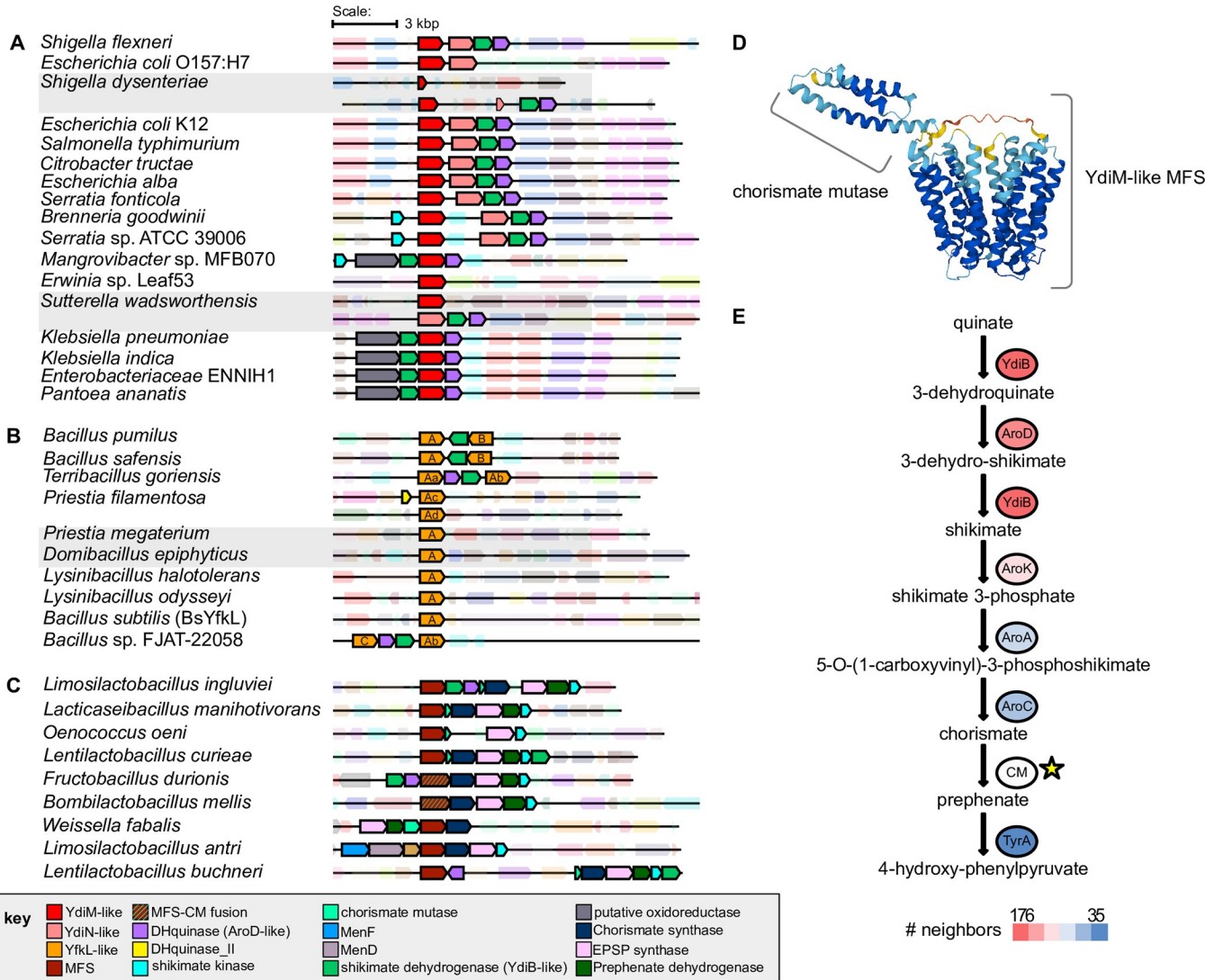

**Fig 6. Phylogenomic analysis of the YdiM/YdiN subfamily.** (**A**) Phylogenetic tree of YdiM, YdiN, and their YdmiM-like and YdiN-like homologues. The taxonomic classification of each leaf, presence of the chorismite mutase fusion, and whether the corresponding gene is next to a gene encoding a shikimate pathway enzyme are indicated with inner rings according to the key shown at the bottom of the figure. Lines colored by taxonomic classification connecting two leaves are used to indicate that those two proteins are encoded by the same genome. The innermost grey ring corresponds to the clusters depicted in panel B, and YfkL indicates the homologue present in *Bacillus subtilus* (Bs). Gene neighborhoods from clades with background shading are shown in Fig 6. Sequences are the 250 best hit from blastp against UniProt reference proteomes. The 10 most similar proteins to YdiM in *E. coli*, *Clostridioides difficile*, *Klebsiella pneumoniae*, *Bacillus subtilis* were used as an outgroup to root the tree. (**B**) Sequence similarity network (SSN) of YdiM/YdiN homologues. Nodes are colored by taxonomy according to the key shown at the bottom of the figure, and clusters are labeled as in the innermost grey ring of panel A. Edge-weighted Spring Embedded Layouts using % id for clustering. (**C**) SSN as in panel B but colored based on presence of neighbor gene(s) encoding enzyme(s) in the shikimate pathway, and (**D**) shows the nodes in red representing the YdiM orthologs with a chorismate mutase fusion.

Firmicutes and ~ 23% are from Proteobacteria. Remarkably, ~ 75% of YdiM/YdiN homologues analyzed here are encoded by a gene that is adjacent (on either the 5' or 3' side) to a gene encoding an enzyme in the shikimate pathway (**Fig 6**). Moreover, this frequency increases to ~ 86% when a larger (20 instead of the usual 10) genes window is used, showing a clear link between the YdiM and YdiN subfamily of the MFS-type transporters and the enzymes of the shikimate pathway (**Fig 7**). In addition to *E. coli*, other Enterobacteriaceae genomes including *Shigella flexneri*, *Salmonella typhimurium*, and *Citrobacter tructae* also

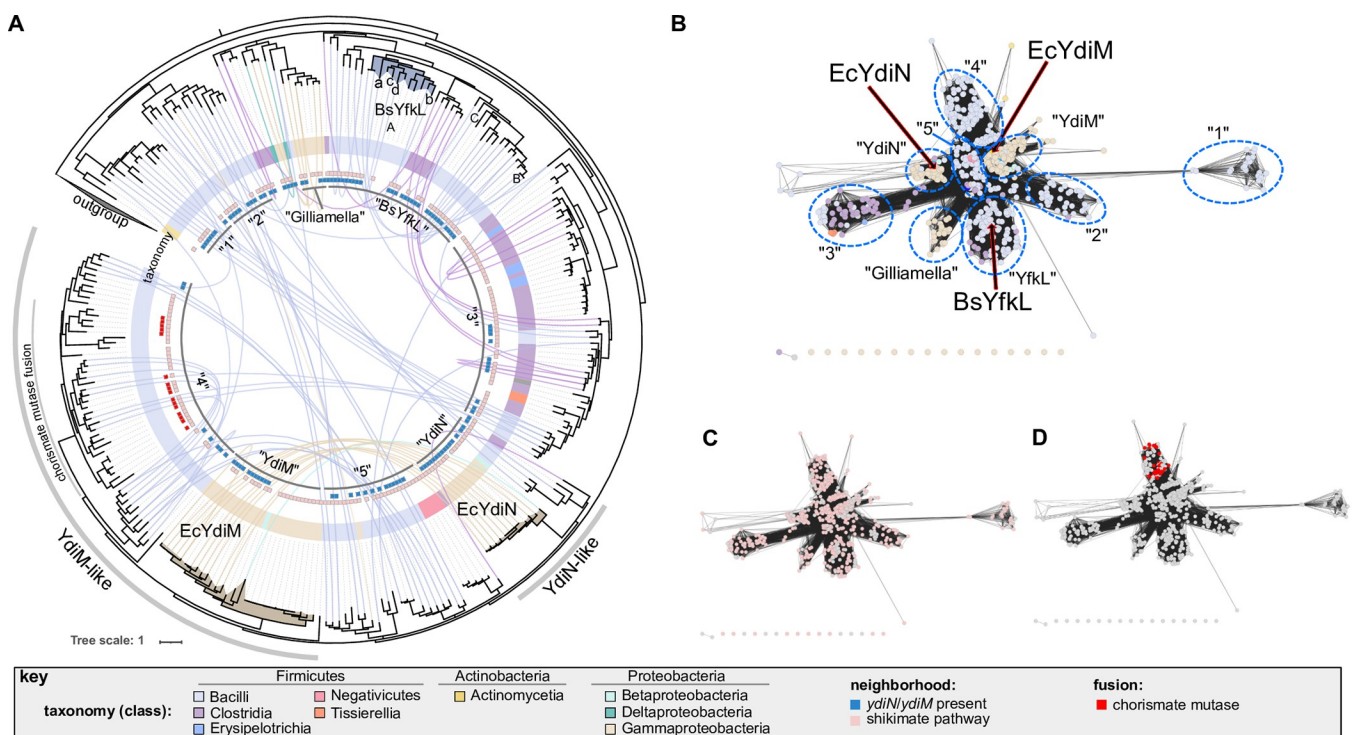

**Fig 7. Gene neighborhoods containing *ydiM/ydiN* homologues.** (**A**) Gene neighborhoods of the YdiM and (**B**) gene neighborhoods of the YdiN clades. (**C**) Representatives from Cluster 4 shown in Fig 5. (**D**) Alphafold prediction depicting the YdiM-chorismate mutase fusion from *Fructobacillus durionis*, showing its two distinct domains. (**E**) Shikimate pathway from quinate to 4-hydroxy-phenylpyruvate and the structural genes of the enzymes involved. A star indicates the step catalyzed by chorismite mutase (CM) that is sometimes found fused to YdiM as shown in (D). The number of times a YdiM/YdiN homologue is encoded in a gene neighborhood with a gene encoding the indicated enzyme is shown as a heatmap (176 red to 35 blue).

encode both YdiM and YdiN located next to the shikimate pathway genes *ydiB* and *aroD* (**Figs 6** and **7**). However, *Klebsiella pneumoniae* encodes only YdiM, and in such Enterobacterial genomes that lack a YdiN ortholog, *ydiM* is found between *aroD* and *ydiB* in a putative operon (**Fig 7A**), further linking YdiM with the shikimate pathway. Moreover, in multiple Firmicutes *ydiM* genes encoding YdiM orthologs are often physically located next to genes encoding shikimate pathway enzymes (**Fig 7B**), and even in a handful of cases, the YdiM orthologs are fused to chorismate mutase (**Fig 7C and 7D**). Chorismate mutase is one of the seven enzymes that form the shikimate pathway (**Fig 7E**) and is often found fused to other enzymes of this pathway and thought to serve regulatory purposes [76]. Of the genes analyzed here, *ydiM* and *ydiN* homologues are most often near the enzymes catalyzing the early steps of the shikimate pathway, starting from quinate (*i.e.*, *ydiB>aroD>aroK*) compared to others operating in the pathway (**Fig 7E**). The overall findings indicated that the YdiM-like proteins are closely associated with the shikimate pathway, and consistent with YdiM and YdiN performing different transport function(s) related to the aromatic acid synthesis pathway.

## Bioinformatic analysis of YfcJ-like proteins

Currently little is known about YfcJ (TC: 2.A.1.46.6) and its homologues. The closest related protein with some associated experimental data is YhhS (TC: 2.A.1.46.7), a paralog of YfcJ in *E. coli*, which was previously linked to cellular arabinose levels [77] and glyophosate (inhibitor of 5-enolpyruvylshikimate-3-phosphate synthase) resistance [78] based on loss-of-function and gain-of-function experiments, respectively. The sequence similarity network and

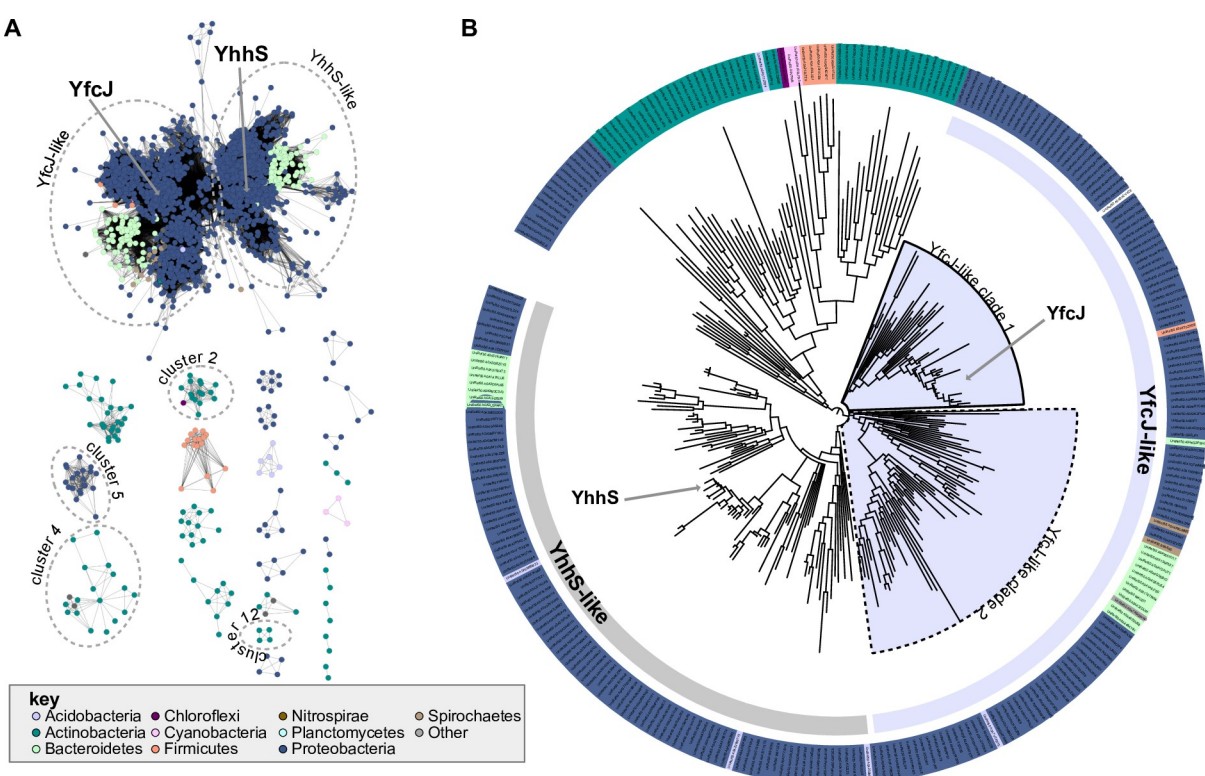

**Fig 8. Sequence similarity analysis of YfcJ homologues.** (**A**) Sequence similarity network using an alignment score of 80. Sequences for the network were collected by searching against UniProt using YfcJ as a query. Clusters with examples of gene proximity in Fig 9 are circled and labeled. The taxonomic classification of each node is colored according to the key shown at the bottom left of the figure. (**B**) Protein sequences from the network were mapped to UniRef50 and representative nodes were used to build a phylogenetic tree. The background color of each leaf is colored according to the key. In addition to clear separation from the YhhS-like clade, the YfcJ group can be distinguished into two major clades, indicated as YfcL-like clade 1 and clade 2.

phylogenetic reconstruction analyses were able to distinguish the YfcJ-like homologues from the closest subfamily composed of YhhS homologues. (**Fig 8A** and **8B**). The YfcJ-like subfamily was mainly identified in Proteobacteria and Bacteroidetes (89% and 8.5% of the homologs, respectively), and within the Proteobacteria, there was a roughly equal split between gamma- (30%), alpha- (30%), and beta- (27%) proteobacteria. Analysis of conserved gene proximity revealed multiple putative operons encoding YfcJ- and YhhS-like transporters. Although defined biochemical functions could readily predicted for proteins encoded by genes neighboring the *yfcJ* homologues, such as amidohydrolases or tautomerases, no specific pathway or process that may be associated with YfcJ could be predicted (**Fig 9**, upper part). Note that the clusters 4, 5, and 12 of YhhS homologues are in putative operons with nucleotide metabolism and tRNA-related proteins, linking the YhhS family to nucleotide metabolism and tRNA modification processes based on conserved gene proximity (**Fig 9**, lower part). Of these clusters, the genes in cluster 4, which is dominated by Actinobacteria, are often in a putative operon with a YacP-like endoribonuclease, and a protein resembling an epoxyqueuosine reductase responsible for a synthesis of queuosine found in some tRNAs. Cluster 5 from Proteobacteria is found in a putative operon with proteins involved in nucleotide metabolism, including a putative hydrolase from the YjjG superfamily involved in cleaving nucleotides with non-canonical nucleotide bases. In cluster 12 the YfcJ- and YhhS-like homologues are in a putative operon with glutamyl-Q tRNA (Asp) synthetase, which is a protein that functions immediately

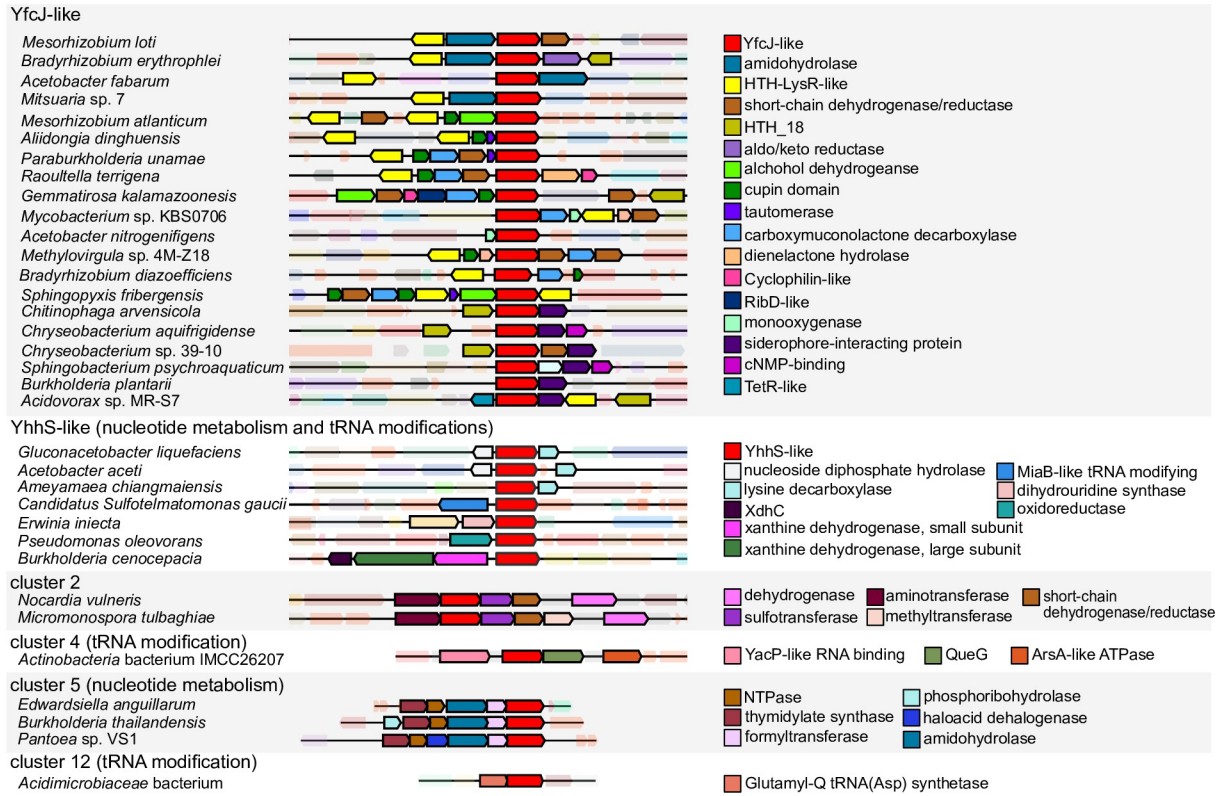

**Fig 9. Conserved gene proximity analysis of YfcJ homologues.** Examples of conserved gene neighborhoods encoding proteins from the YfcJ sequence similarity network are shown. Clusters corresponding to genes involved in nucleotide metabolism and tRNA modification (YhhS-like). tRNA modification (cluster 4), nucleotide metabolism (cluster 5) and tRNA modification (cluster 12) are shown with the related enzymes found shown on the right.

downstream of epoxyqueuosine reductase involved in the synthesis of the hypermodified base glutamyl-queuosine [79]. For cluster 2 in the network, which is largely confined to Actinobacteria, all YfcJ- and YhhS-like homologues are encoded by a gene that is potentially in a biosynthetic gene cluster for an unknown secondary metabolite, suggesting that they may be metabolite transporters.

## Discussion

The MFS-type transporter CcoA (TC: 2.A.1.81.1) of the CalT subfamily is a well-established Cu importer identified in bacteria and required for biogenesis of the *cbb*$_3$-Cox Cu$_B$ center [11,46,49]. However, although CcoA is widespread among alpha-proteobacterial species and it frequently co-occurs with the genes encoding *aa*$_3$-Cox and *cbb*$_3$-Cox [48,49], it is only required for *cbb*$_3$-Cox and not the quasi-identical Cu$_B$ center containing *aa*$_3$-Cox, as seen with *R. sphaeroides* [49]. Moreover, no functional ortholog of *R. capsulatus* CcoA is found among the ~ 70 MFS-type transporter genes of *E. coli*, suggesting a different mechanism for Cu import and biogenesis for the *bo*$_3$-Cox Cu$_B$ center. These observations point out the specificity of the CalT members among the MFS-type transporters and indicate the possible occurrence of different routes for the biogenesis of Cu$_B$ centers of heme-Cu enzymes (*e.g.*, *E. coli bo*$_3$-Cox). Indeed, bacterial *cbb*$_3$-Cox and *aa*$_3$-Cox require specific transporters and chaperones for the biogenesis of their Cu$_B$ centers assembly, including the periplasmic Sco-like [31,35] and PCuAC-like chaperones [2,32,33,38]. Moreover, *cbb*$_3$-Cox requires in addition to the Cu

importer CcoA [46] the cupric reductase CcoG [43], and the $P_{1B}$-type transporter CcoI/CtpA [26,44,45]. Remarkably, none of the latter proteins are involved in the case of the $aa_3$-Cox, which instead uses the Cu chaperone Cox11 [36–38]. How the $Cu_B$ center insertion occurs in *E. coli* $bo_3$-Qox is not known, and as a true CalT homologue does not seem to exist in this species, raising the issue of whether any other type of MFS-transporter might accomplish this function.

A survey of the *E. coli* genome indicated that among the ~ 70 MFS-type transporters, ~ 28 of them (*i.e.*, UMFs) had no identified cargo, and at least eight of them were richly endowed with plausible metal-coordinating amino acid residues. This enticed us to examine the role of these UMFs in $bo_3$-Cox biogenesis, using a genetic screen based on the essentiality for aerobic respiratory growth sustained by this enzyme in the absence the *bd*-Qox1 and *bd*-Qox2. This screen identified YhjE, YdiM, and YfcJ as required MFS-type transporters for $bo_3$-Cox dependent aerobic respiratory growth of *E. coli*. In the absence of any one of these proteins, the $bo_3$-Qox activity and its *b*- and o-type hemes were absent, even though at low cell-densities detectable amounts of *cyoABCD* mRNA transcripts were produced. Remarkably, a multicopy plasmid carrying these genes and overproducing the $bo_3$-Qox could bypass at least partially the need for these UMFs. These findings suggested that some regulatory event(s) (*e.g.*, titrating out a regulator) controlling the transcription or destabilizing the transcript(s) might occur in the absence of these UMFs. Alternatively, although these mutants might produce the structural constituents of the $bo_3$-Qox, they could not assemble an active enzyme in the absence of the imported/exported cargo(s). Thus, the specific nature(s) of currently unidentified cargos transported by these MFS-type transporters seem important for $bo_3$-Qox biogenesis. Earlier genetic studies have suggested that *yhjE*, *ydiM*, and *ycfJ* may be involved in transporting an unknown metabolite (see TCDB), isoprenol [75] and arabinose [77], respectively. Here, the whole-cell transport assays further showed that cells without YdiM accumulated less $^{64}Cu$, and those without YhjE contained more reduced $^{55}Fe$ (**Fig 4**), while no such difference was seen in the absence of YfcJ. Note that currently no conclusive data exist for any of these transporters, as none of them has been purified and shown to bind and transport their putative substrates.

Bioinformatics analysis have been performed to unravel the function of YfcJ, YhjE and YdiM. No link between $bo_3$-Qox and metal transport were found for YfcJ. YhjE was referred to as a member of the metabolite: H+ symporter (MHS) Family (see TCDB), and phylogenomic analyses show that in many bacterial genomes, *yhjE* gene clusters with the $bo_3$-Qox structural genes *cyoABCD*, suggesting that the role of YhjE-like transporters in $bo_3$-Qox function could be widely conserved (**Fig 7**). Based on an analysis of previously published high-throughput (HTP) interaction data [80], YhjE was found to physically interact with FhuA, a ferrichrome outer membrane transporter [81]. Out of 331 identified genetic interactions in a separate study, a positive genetic interaction was identified with *fhuA* (*i.e.*, the double *yhjE fhuA* mutant grew better in rich medium than the single mutants), and negative genetic interactions with other Fe transporters (*fecA*, *fecB*, *fecC*, *fecD*, *fepA*, *fepB*, *febD*, *fes*, and *fhuC*) [80]. If these results obtained with HTP studies are not misleading false positives, they could potentially explain the Fe-homeostasis defect in the Δ*yhjE* strain. A negative genetic interaction was also observed between *yhjE* and *cyoA* or *cyoB* and a positive genetic interaction with *cyoC*. Such results may suggest that YhjE could have functional roles beyond $bo_3$-Qox biogenesis. Overall, the available experimental and bioinformatic data support that this transporter is required to produce an active $bo_3$-Qox, but the underlying molecular link(s) remains unknown.

YdiM was initially selected as a candidate metal transporter based on the presence of $M_{21}XXXXM_{26}$ and $M_{76}XXM_{79}XXXM_{83}$ motifs in its predicted TM1 and TM3, and other motifs in the TM4, and TM6 (**S5 Fig**). The 3D structural model of YdiM is reminiscent of that of CcoA since the Met residues are positioned in a similar fashion throughout the TMs of both

proteins. However, the putative transmembrane metal-binding motif(s) are different from the CalT subfamily members (**S5 Fig**) [11]. These putative metal binding residues combined with the experimental data presented here suggest that YdiM could be a plausible candidate for Cu transport. In the *E. coli* K-12 genome *ydiM* gene and its paralog *ydiN* are clustered together with several genes involved in the shikimate pathway, which is the metabolic pathway governing biosynthesis of aromatic amino acids, like phenylalanine, tyrosine, and tryptophan [82,83]. Phylogenetic analyses of bacterial species other than *E. coli* also indicate that *ydiM* and *ydiN* cluster frequently with the shikimate pathway genes (**Figs 5** and **6**). An earlier study indicated that the 3-deoxy-D-arabino-hepulosonate-7-phosphate synthase (DAHP synthase) catalyzing the first step of this pathway binds Cu, suggesting that the DAHP synthase may be a cuproenzyme [84]. However, no conclusive study has been conducted, leaving the identity of the metal of DAHPS contested [85], and the link between Cu, the shikimate biosynthetic pathway, and $bo_3$-Qox remains unclear, deserving future studies.

In summary, this study unexpectedly implicated three MFS-type transporters, YhjE, YdiM and YfcJ in the production of an active $bo_3$-Qox in *E. coli*. Available data showing impaired Cu and Fe uptake kinetics suggest that YdiM and YhjE are involved in cellular metal homeostasis, which may be essential for the biogenesis of the heme-Cu enzyme $bo_3$-Cox. However, the cargo of these transporters being currently unknown, and their role(s) in specific metabolic pathway(s) undefined, a direct mechanistic link between them and the expression or assembly of the $bo_3$-type Qox remains hypothetical until such data become available. Nonetheless, the overall findings increased the arsenal of the different gene products that cells use to produce heme-Cu enzymes, including the $bo_3$-Qox. These studies also illustrated how broad a biological function the MFS-type transporters may play in cells and spur future investigations to identify the transported substrates and shed light to the mechanistic link(s) between these MFS-type transporters and the biogenesis of heme-Cu containing metalloproteins.

## Supporting information

**S1 Fig.**
(PDF)

**S2 Fig.**
(PDF)

**S3 Fig.**
(PDF)

**S4 Fig.**
(PDF)

**S5 Fig.**
(PDF)

**S1 File.**
(DOCX)

**S1 Dataset.**
(XLSX)

## Acknowledgments

We thank Drs. R. B. Gennis for the plasmid pJRHisA(cyoABCD), A. Dancis for help with performing Fe uptake kinetics, and M. Goulian for providing *E. coli* strains, phage, and plasmids.

## Author Contributions

**Conceptualization:** Crysten E. Blaby-Haas, Fevzi Daldal.

**Data curation:** Fevzi Daldal.

**Formal analysis:** Bahia Khalfaoui-Hassani, Crysten E. Blaby-Haas, Andreia Verissimo.

**Funding acquisition:** Crysten E. Blaby-Haas, Fevzi Daldal.

**Investigation:** Bahia Khalfaoui-Hassani, Fevzi Daldal.

**Methodology:** Andreia Verissimo.

**Visualization:** Crysten E. Blaby-Haas.

**Writing – original draft:** Bahia Khalfaoui-Hassani, Crysten E. Blaby-Haas.

**Writing – review & editing:** Bahia Khalfaoui-Hassani, Crysten E. Blaby-Haas, Andreia Verissimo, Fevzi Daldal.

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
