## [Decision Letter · Decision Letter 0]

24 Aug 2023

PONE-D-23-23379The Escherichia coli MFS-type transporter genes yhjE, ydiM, and yfcJ are required to produce an active bo3 quinol oxidasePLOS ONE

Dear Dr. Daldal,

Thank you for submitting your manuscript to PLOS ONE. After careful consideration, we feel that it has merit but does not fully meet PLOS ONE’s publication criteria as it currently stands. Therefore, we invite you to submit a revised version of the manuscript that addresses the points raised during the review process.

Please submit your revised manuscript by Oct 08 2023 11:59PM . If you will need more time than this to complete your revisions, please reply to this message or contact the journal office at plosone@plos.org. Please include the following items when submitting your revised manuscript:A rebuttal letter that responds to each point raised by the academic editor and reviewer(s). You should upload this letter as a separate file labeled 'Response to Reviewers'.A marked-up copy of your manuscript that highlights changes made to the original version. You should upload this as a separate file labeled 'Revised Manuscript with Track Changes'.An unmarked version of your revised paper without tracked changes. You should upload this as a separate file labeled 'Manuscript'.

We look forward to receiving your revised manuscript.

Kind regards,

Tarunendu Mapder, Ph.D.

Academic Editor

PLOS ONE

Journal Requirements:

   "This work was supported by DOE grant DE-FG02-91ER20052 (FD). Work at the Molecular Foundry was supported by the Office of Science, Office of Basic Energy Sciences, of the U.S. Department of Energy under Contract No. DE-AC02-05CH11231 (CEB-H). Work at the U.S. Department of Energy Joint Genome Institute (https://ror.org/04xm1d337), a DOE Office of Science User Facility, is supported by the Office of Science of the U.S. Department of Energy operated under Contract No. DE-AC02-05CH11231 (CEB-H). "

7. PLOS ONE now requires that authors provide the original uncropped and unadjusted images underlying all blot or gel results reported in a submission’s figures or Supporting Information files. This policy and the journal’s other requirements for blot/gel reporting and figure preparation are described in detail at https://journals.plos.org/plosone/s/figures#loc-blot-and-gel-reporting-requirements and https://journals.plos.org/plosone/s/figures#loc-preparing-figures-from-image-files. When you submit your revised manuscript, please ensure that your figures adhere fully to these guidelines and provide the original underlying images for all blot or gel data reported in your submission. See the following link for instructions on providing the original image data: https://journals.plos.org/plosone/s/figures#loc-original-images-for-blots-and-gels. 

Reviewers' comments:

Reviewer's Responses to Questions

**Comments to the Author**

1. Is the manuscript technically sound, and do the data support the conclusions?

Reviewer #1: Yes

Reviewer #2: Yes

2. Has the statistical analysis been performed appropriately and rigorously? 

Reviewer #1: Yes

Reviewer #2: I Don't Know

3. Have the authors made all data underlying the findings in their manuscript fully available?

Reviewer #1: Yes

Reviewer #2: Yes

4. Is the manuscript presented in an intelligible fashion and written in standard English?

Reviewer #1: Yes

Reviewer #2: Yes

5. Review Comments to the Author

Reviewer #1: Journal: PLOS ONE

Manuscript Type: Research Article

Manuscript ID: PONE-D-23-23379

Title: The Escherichia coli MFS-type transporter genes yhjE, ydiM, and yfcJ are required to

produce an active bo3 quinol oxidase

The current work by Khalfaoui-Hassani et al investigates molecular insights into the production of an active bo3 quinol oxidase in Escherichia coli. The group identified a subset of uncharacterized MFS transporters, based on the presence of putative metal-binding residues, as likely candidates for the missing Cu transporter. Phylogenomic analyses also suggest plausible roles for the YhjE, YdiM, and YfcJ transporters, and overall findings illustrate the diverse roles that the MFS-type transporters play in cellular metal homeostasis and production of active heme-Cu oxygen reductases. Overall evaluation shows that the study has been well executed and the results are also well presented. The conclusion of the study is substantiated by the experimental outcomes. The authors have listed the limitations of their study with the future scope/directions. There are a few minor suggestions. If rectified, the article will be suitable for publication in PlosOne.

1. The authors should elaborate the introduction with the current literature support.

2. The limitations and prospects of the study should be elaborated more for the readers.

Reviewer #2: Khalfaoui-Hassani et al. discovered eight uncharacterized MFS-type transporters (YfcJ, YhjX, YebQ, YnfM, YdiM, YhjE, AraJ, and SetC) containing potential copper-binding motifs. YhjE, YdiM, and YfcJ play a crucial role in active bo3-Qox enzyme production during E. coli aerobic growth. Mutants lacking these transporters showed defects in aerobic growth and bo3-Qox activity, suggesting their importance in bo3-Qox biogenesis. YhjE and YdiM appear to be involved in copper and iron homeostasis, respectively, as their mutants exhibited altered metal uptake. Bioinformatic analyses provided insights into the potential roles of these transporters in known pathways. This research advances our understanding of heme-copper enzyme biogenesis and cellular metabolism. The manuscript is well written and easy to read.

Comments:

1. The paper does not elucidate the specific mechanism by which these MFS-type transporters are involved in bo3-Qox biogenesis. The authors propose that they might play a role in regulating gene expression or stabilizing mRNA transcripts, but this remains speculative. This More detailed mechanistic studies are required to flesh out their specific functions in bo3-Qox production.

2. How are the YhjE, YdiM, and YfcJ transporters involved in copper and iron homeostasis? Is there any evidence suggesting their direct interactions with these metals?

3. The study suggests that YhjE and YdiM are related to the shikimate pathway. How might their involvement in this pathway be linked to their roles in bo3-Qox biogenesis?

4. The authors should add “Statistical analyses” in the Materials and Methods. The figure legends should describe the errors bars and N numbers (Ex: Fig. 4)

6. PLOS authors have the option to publish the peer review history of their article (what does this mean?). If published, this will include your full peer review and any attached files.

Reviewer #1: **Yes: **Manishekhar Kumar

Reviewer #2: **Yes: **Jagannath Misra

---

## [Author Response · Author response to Decision Letter 0]

13 Sep 2023

see the revised cover letter and the Responses to the Reviewers files uploaded

---

## [Editor Report · Decision Letter 1]

4 Oct 2023

The Escherichia coli MFS-type transporter genes yhjE, ydiM, and yfcJ are required to produce an active bo3 quinol oxidase

PONE-D-23-23379R1

Dear Dr. Daldal,

We’re pleased to inform you that your manuscript has been judged scientifically suitable for publication and will be formally accepted for publication once it meets all outstanding technical requirements.

Kind regards,

Tarunendu Mapder, Ph.D.

Academic Editor

PLOS ONE
---

## [Editor Report · Acceptance letter]

11 Oct 2023

PONE-D-23-23379R1 

The *Escherichia coli* MFS-type transporter genes *yhjE, ydiM, and yfcJ* are required to produce an active *bo*_3_ quinol oxidase 

Dear Dr. Daldal:

I'm pleased to inform you that your manuscript has been deemed suitable for publication in PLOS ONE. Congratulations! Your manuscript is now with our production department. 

Kind regards, 

on behalf of

Dr. Tarunendu Mapder 

Academic Editor

PLOS ONE